# Towards Understanding the Shape of Representations in Protein Language Models

**Kosio Beshkov**
Department of Physics
University of Oslo
Oslo, Norway
kosio.neuro@gmail.com

**Anders Malthe-Sørenssen**
Department of Physics
University of Oslo
Oslo, Norway

## Abstract

While protein language models (PLMs) are one of the most promising avenues of research for future de novo protein design, the way in which they transform sequences to hidden representations, as well as the information encoded in such representations is yet to be fully understood. Several works have attempted to propose interpretability tools for PLMs, but they have focused on understanding how individual sequences are transformed by such models. Therefore, the way in which PLMs transform the whole space of sequences along with their relations is still unknown. In this work we attempt to understand this transformed space of sequences by identifying protein structure and representation with square-root velocity (SRV) representations and graph filtrations. Both approaches naturally lead to a metric space in which pairs of proteins or protein representations can be compared with each other.

We analyze different types of proteins from the SCOPe dataset and show that the Fréchet radius and effective dimension of the SRV shape space follow a non-linear pattern as a function of the layers in ESM2 models of different sizes. Furthermore, we use graph filtrations as a tool to study the context lengths at which models encode the structural features of proteins. We find that PLMs preferentially encode immediate as well as local relations between residues but start to degrade for larger context lengths. The most structurally faithful encoding tends to occur close to, but before the last layer of the models, indicating that training a folding model on top of these layers might lead to improved folding performance.

## 1 Introduction

Protein language models (PLMs) are a novel and powerful approach for the modeling and design of new proteins with desired structural or functional features (Ferruz & Höcker, 2022). Using PLMs, one can predict the folding of proteins in 3D space (Lin et al., 2023), generate new candidate sequences for viral vectors (Lyu et al., 2024), enzymes (Madani et al., 2023), biosensors (Hayes et al., 2025) and functional binders (Chen et al., 2025; Bryant & Elofsson, 2023). A particular setting in which PLMs have been found to be useful is in finding high-dimensional representations of protein sequences that are thought to reflect the physical, evolutionary, or functional properties of a protein. This is especially useful since it allows one to efficiently evaluate and compare newly generated proteins without the need for expensive modeling or experiments. One of the most widespread models of this type are the Evolutionary Scale Modeling (ESM) models (Lin et al., 2023; Hayes et al., 2025), which we will analyze in this work.

It is empirically understood that the representations in these models form a good initialization for folding models (Lin et al., 2023) and can also be used as a reward function to guide other generative approaches (Wang et al., 2025). However, the precise features that these representations encode are still not fully understood. In a previous work by Zhang et al. (2024), the authors develop a categorical Jacobian approach to suggest that PLMs encode the pairwise statistics of coevolving residues. In another work by Simon & Zou (2024), the authors use sparse autoencoders to suggest that PLMs encode human-interpretable features such as binding, structural motifs and functional domains. In

yet another work that leverages the power of sparse autoencoders (Gujral et al., 2025), the authors show that PLMs and the individual neurons in them also encode features related to biologically relevant terms within the Gene Ontology hierarchy (Aleksander et al., 2023).

What all of these works study is the transformation of individual sequences to high-dimensional latent representations. However, such approaches ignore how different proteins or their representations relate to each other in the latent space of a PLM. If structure determines function, then it is reasonable to assume that similar structure determines similar function. Furthermore, if protein representations in PLMs characterize structural, evolutionary and functional features, then one might expect that similar representations share such features. In addition, the representation of a protein in a PLM is a tensor of shape *amino acids×model dimension*, however for many applications it is standard to take the average over the first dimension, which ignores the shape of the representation, thereby missing the full richness of the information present in PLM representations.

To state our motivation more formally, it is important to understand how the metric space of proteins compares to the metric space of complete shape representations in PLMs. Metric space approaches for the structural analysis of proteins have previously been considered within the rich field of shape analysis (Liu et al., 2010). The essential realization of these approaches is that two proteins can be structurally compared by finding the optimal way to superimpose them, allowing different types of transformations. This is the backbone for many standard tools in the field of rigid and flexible structural alignment such as root mean square deviation (RMSD), TM score (Zhang & Skolnick, 2004) and FATCAT (Li et al., 2020) among others. Despite their popularity, to our knowledge, such methods have yet to be applied to study the hidden representations of PLMs.

In this work, we adapt and extend the shape-analysis framework to study the layerwise metric representation spaces of eight different classes in the SCOPe (Chandonia et al., 2022) dataset pushed through several ESM2 and Ankh models. We consider different features such as the Fréchet radius and the effective dimensionality of these spaces and show that they follow a peculiar pattern as a function of layer, which is especially prominent for larger models. Furthermore, we introduce a graph-filtration method that allows us to separate and study the scale at which PLMs best maintain the structure of proteins. Using this analysis, we show that while structure is always encoded better than chance in deep PLM representations, it is optimally encoded at very short context lengths of 2 or at slightly longer context lengths at $\sim 8$ amino-acid neighbors.

## 2 BACKGROUND

In this section, we fix our notation and introduce the mathematical machinery necessary for comparing proteins and their corresponding representations in protein language models. We define two different ways to compare proteins with PLM representations. In the first one, each protein is a point in a high-dimensional (sometimes infinite-dimensional) space, and we define a metric that can be consistently applied to proteins made up of different numbers of amino acids. In the second approach, we define a filtration of metric spaces that only allows us to compare proteins of the same size. As we will argue, this is useful for understanding the context length that current PLMs are sensitive to, as well as the degree to which protein structure is encoded in PLMs.

We start by outlining several ways to mathematically formulate what proteins are. For each such definition of a "protein" we also propose a metric space in which such proteins can be sensibly embedded.

1. *Proteins as sequences*: In this case, we define a protein $P$ by its amino acid sequence. So, a protein of length $L$ will live in a space $\mathcal{A}^L$, where $\mathcal{A}$ is an alphabet of the 20 canonical amino acids. Given that we want to compare proteins with different numbers of amino acids, we can define the space of all possible amino acid sequences as $\mathcal{A}^* = \bigcup_{L=0}^{\infty} \mathcal{A}^L$. An edit distance, such as the Levenshtein distance for example, can be used to define a metric on this space. Thus, the metric space defined by amino acid sequences will be the pair $(\mathcal{A}^*, d_{lev})$.

2. *Proteins as three-dimensional point clouds*: If we consider the actual physical structure of a protein, we can define a protein as an ordered point cloud of size $L$ in $\mathbb{R}^3$ equipped with the Euclidean metric. However, $\mathbb{R}^3$ contains points rather than point clouds, so point clouds

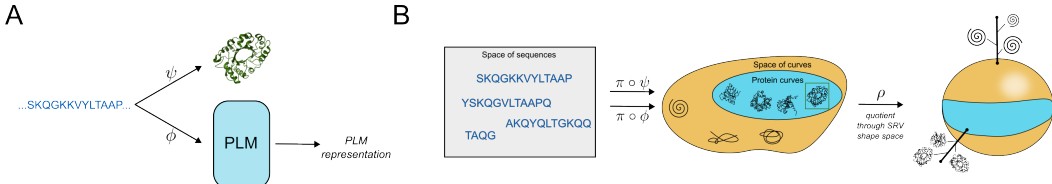

Figure 1: **A**) Depiction of a single sequence (1kr1) mapped to a three-dimensional structure and a PLM representation. **B**) Illustration of how the space of sequences is first transformed into a space of curves and afterwards into a shape space. The lines sticking out of the sphere show the fibers of the map $\rho$.

of arbitrary sizes live in a different space, namely $\mathcal{P}_3^* = \bigsqcup_{n=0}^{\infty} (\mathbb{R}^3)^n$. While it is possible to define a metric on this space, such metrics (for example, Hausdorff or Wasserstein) often ignore the ordering of the point cloud, which matters when studying proteins. Note that each protein can be described through a map $\psi : \mathcal{A}^* \to \mathcal{P}_3^*$ from the sequence space to $L$ points in $\mathbb{R}^3$.

3. *Proteins as curves*: Instead, we can study structure by identifying proteins with curves in $\mathbb{R}^3$. This is done by identifying the space of proteins with the space of continuous curves $\Gamma_3 = \{\gamma : \gamma : [0,1] \to \mathbb{R}^3\}$. We further require that for every protein of length $L$ and coordinates $\psi_l(P) \in \mathbb{R}^3$ we have $\gamma(t) = \psi_l(P)$ if $l = tL \in \mathbb{N}$. As we will discuss later, this definition of a protein treats proteins of any length as the same object and makes it easier to define a metric that compares them.

4. *Proteins as graphs*: Finally, proteins can also often be thought of as graphs. This usually requires a contact map, which is a binary matrix that encodes whether two residues are closer than a specified threshold (6-12Å being a standard choice). We define this approach more rigorously in subsection 2.2.

Notice that while we have defined different ways to think of proteins, the second and third definitions can also be generalized to any ordered point cloud. Therefore, we can use them to study three-dimensional protein structure as well as the embeddings of sequences in protein language models. We will denote the map from an amino acid sequence to the embedding space of a language model by the function $\phi : \mathcal{A}^* \to \mathcal{P}_m^*$, where $m >> 3$ is the embedding dimension of the model. Thus, our approach can be summarized by Figure 1 and the following diagram,

$$
\begin{array}{ccc}
\mathcal{A}^* & \xrightarrow{\phi} & \mathcal{P}_m^* \\
\psi \downarrow & & \downarrow \pi \\
\mathcal{P}_3^* & \xrightarrow{\pi} & \Gamma_m
\end{array}
\tag{1}
$$

where $\pi$ is a map that sends point clouds in different spaces to a shared metric space of curves in $\mathbb{R}^m$. Practically, one can think of $\pi$ as a choice of how to interpolate ordered point clouds so that they become curves in $\mathbb{R}^m$.

## 2.1 SHAPE SPACES AND THE SQUARE-ROOT VELOCITY REPRESENTATION

A fundamental feature of protein structure is invariance to rotations and translations. What this implies is that protein structure is invariant under isometries of $\mathbb{R}^3$ or the special Euclidean group $SE(3)$. This is the principle behind models such as the SE(3)-transformer (Fuchs et al., 2020), which is equivariant to transformations from this group and is thought to play an important role in the success of folding models such as AlphaFold2 (Jumper et al., 2021) and RoseTTAFold (Baek et al., 2021).

In the present work we use the square-root velocity (SRV) framework introduced in Srivastava et al. (2010) to enforce invariance to translation. This approach has previously been used to create a

geometry-aware Variational Autoencoder (VAE) model in Huang et al. (2021). For any ordered point cloud, we interpolate the points with quadratic splines, thereby generating a curve $\gamma$ which has the same length independently of the number of amino-acids in a protein sequence. Following the interpolation step, the SRV representation is defined as,

$$q(t) = \dot{\gamma}(t)/\sqrt{\|\dot{\gamma}(t)\|_2}. \tag{2}$$

From the norm in the denominator one can see that this approach projects curves to an infinite-dimensional sphere $S^\infty$, which makes computing geodesics, and thereby measuring distances, straightforward. Since translations are already accounted for, we quotient out the remaining actions of the $SE(m)$ group, namely the rotations generated by $SO(m)$. This is done using SVD to solve the optimization problem for any two SRV curves $q_1$ and $q_2$,

$$\hat{R} = \underset{R \in SO(n)}{\arg\min} \|q_1 - Rq_2\|_2. \tag{3}$$

Given this, the distance between any two curves is defined in terms of the L2 norm as $d(q_1, q_2) = \|q_1 - \hat{R}q_2\|_2$. It is often the case that one also removes different reparameterizations of curves, but we have avoided that step given that interpolating between residues forms a consistent way of parameterizing proteins, and this additional step is not worth the computational cost.

In summary, we map sequences to their three-dimensional structure and to their representations in a PLM. These form point clouds which we interpolate to map to curves that are then transformed to their SRV representation. Finally, we quotient out rotations to form the shape space, which inherits a Riemannian structure and allows for the efficient computation of distances. We denote this procedure by a quotient map $\rho : \Gamma_m \to \Gamma_m/SE(m)$. Notice that the space of protein curves under this map form a submanifold of the quotient space $H = S^\infty/SO(m)$ as illustrated in panel **B** of Figure 1. All curves that are the same up to actions of $SE(m)$ end up at a single point $y$ on this submanifold. The set of these curves is called a fiber and is defined by $\rho^{-1}(y) = \{\gamma|\rho(\gamma) = y\}$.

## 2.2 Graph filtrations

When making predictions, language models have to consider the context within which a token exists. The context length to which a model is sensitive is unknown a priori. Furthermore, while using a contact map with a threshold of 6-12Å makes sense in real three-dimensional protein structure, it is less clear how to choose such a threshold in PLM representations where such a unit is not defined. For this reason, one needs to work with methods that are capable of considering many possible context lengths. One such example is the concept of a filtration, which is fundamental in the topological data analysis literature (Edelsbrunner & Harer, 2010) and has been extended to graphs within the field of graph learning (Hofer et al., 2020; O'Bray et al., 2021).

In a filtration, one defines a parameter $t$ and a family of objects $S_t \subset S_{t'}$ whenever $t < t'$. In this specific case, we consider filtrations of graphs composed of a set of vertices $V = \{v_1, v_2, ..., v_N\}$ with coordinates $v_i \in \mathbb{R}^m$ and edges whenever $d_2(v_i, v_j) < t$, with $d_2$ being the Euclidean metric enforced by the ambient space. This induces a filtration on their adjacency matrices $A^t$, since $A_{i,j}^t \leq A_{i,j}^{t'}$ for all $i, j$ when $t < t'$.

Comparing the coordinates of real proteins in $\mathbb{R}^3$ to those of a PLM in $\mathbb{R}^m$ is not a well-defined procedure. However, for the same protein $P$ of length $L$, both adjacency matrices live in $\{0, 1\}^{L \times L}$. This allows us to compare the structure of a protein with the structure of its embedding in a PLM. A natural choice of a metric in this space is the entry-wise 1-norm of the difference between the two adjacency matrices or $d_A(\psi(P), \phi(P)) = \|\psi(A^t) - \phi(A^t)\|_1$, where by abuse of notation $\psi(A^t)$ and $\phi(A^t)$ respectively indicate the flattened adjacency matrix of the three-dimensional structure and the PLM representation at the $t$-th filtration value.

For our analysis we construct the $k$ nearest neighbor graphs of the true protein structure and PLM representation as shown in Figure 2. For small $k$, the graphs can only differ at a few locations, and the distance between their adjacency matrices is small. On the other hand, as one increases the number of neighbors, the graphs converge to cliques, and therefore this distance converges to 0. In

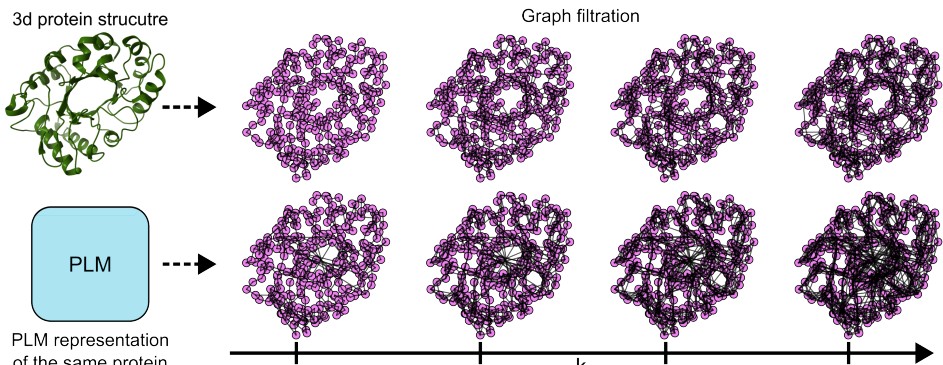

Figure 2: Illustration of how each protein or PLM representation is transformed to a filtration of graphs with a different number of neighbors. The connectivity in the PLM representation is superimposed over the three-dimensional protein graph for clarity.

general, these distances tend to follow a hypergeometric distribution over $k$. To counteract this, we normalize the distance by the empirical distribution of distances between real proteins and random point clouds $R_i \subset \mathbb{R}^m$. Therefore, for a family of proteins $\mathbf{P} = \{P_1, P_2, ..., P_N\}$, we get a filtration of distance histograms $\{d_A(\psi(P_i), \phi(P_i))\}_i^k$. At each level of the filtration we study the normalized first moment, which we will refer to as the *graph filtration moment*. It is given by the equation

$$\mathbb{E}_{P_i \in \mathbf{P}}[d(P_i, \phi(P_i))]_{norm} = \frac{\mathbb{E}_{P_i \in \mathbf{P}}[d_A(\psi(P_i), \phi(P_i))]}{\mathbb{E}_{P_i \in \mathbf{P}, R_i \in \mathbf{R}}[d_A(\psi(P_i), R_i)]}. \quad (4)$$

## 3 RESULTS

All of the following analysis is based on 1377 randomly sampled protein structures from the SCOPe dataset (Chandonia et al., 2022). We sampled up to 200 proteins (some proteins were excluded due to missing a pdb file) from each of the following protein classes – [Alpha, Beta, Alpha/Beta, Alpha+Beta, Alpha and Beta, Membrane and cell surface proteins and peptides, Small proteins and Designed proteins]. More information about these protein classes can be found on the SCOPe website https://scop.berkeley.edu/ or in Chandonia et al. (2022), the exact proteins used in this work along with the code for the analysis can be found in our Github repository https://github.com/KBeshkov/ProtGeom/.

### 3.1 GEOMETRY OF PLM SHAPE SPACES

In section 2.1 we defined a notion of a shape space that inherits a Riemannian structure that allows us to compute distances and also carry out computations in the tangent space of each fiber. With this in hand, we can define and estimate statistics of curves on the shape space as well as make use of tools such as tangent PCA. We therefore track two measures of PLM shape space geometry, namely effective dimension and Fréchet radius. For all computations we use the Geomstats package (Miolane et al., 2020).

For a set of proteins $\mathbf{P} = \{P_1, P_2, ..., P_N\}$, we first interpolate them at 1000 equally spaced points with quadratic splines and then project them to the SRV shape space giving us the set $\mathbf{Y} = \{y_1, y_2, ..., y_N\}$. There is not an a priori correct choice of spline order, but quadratic splines are the simplest option that is still differentiable as required by the SRV framework. Very high-order splines are expected to add a lot of additional structure to the PLM point clouds and could thus bias the results. For an evaluation of the robustness of our results to spline order and interpolation samples see Figure 6. We compute the Fréchet radius by finding the Fréchet mean $p_F = \arg\min_{x \in H} \sum d(x, y_i)$

through gradient descent after which the Fréchet radius is defined as,

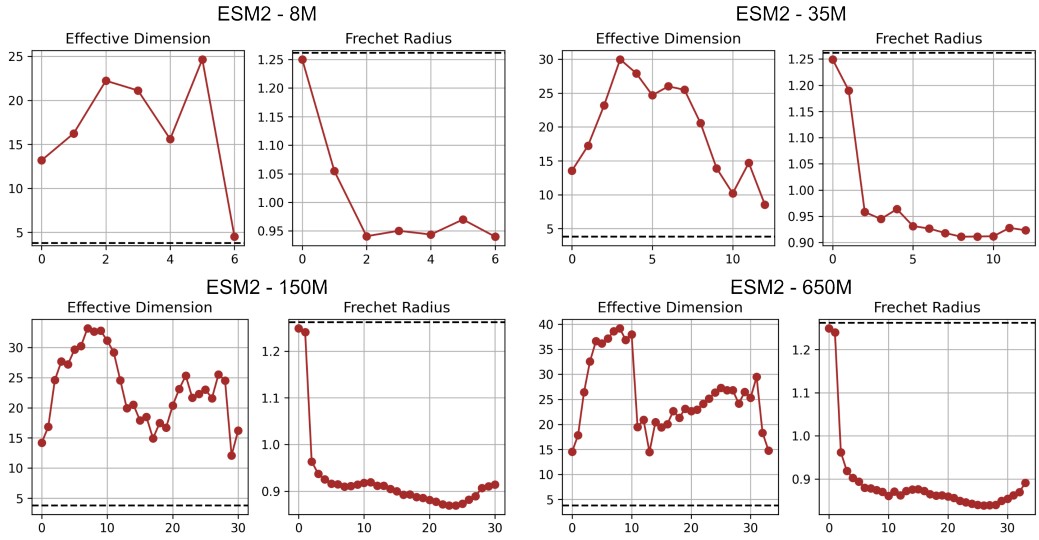

Figure 3: Effective dimension and Fréchet radius for each of the four models as a function of the layers. The black line indicates the value of each metric for the three-dimensional protein structure. As one can see larger models exhibit dimension expansion in the initial layers and contraction later on.

$$r_F = \mathop{\mathbb{E}}_{y_i \in \mathbf{Y}} [d(y_i, p_F)]. \tag{5}$$

An intuitive way to think of this object is as a measure for how spread out PLM representations are with respect to each other on the shape space. Therefore, a small value indicates that different proteins are represented by similar shapes, whereas a larger value indicates the opposite. As one can see in Figure 3 the Fréchet radius tends to decrease with depth and is much smaller for PLM representations than for real three-dimensional protein structures. Surprisingly enough, it does not seem to vary with model size, indicating that the variability among shapes is low for all models.

Another measure describing the geometry of shape spaces in PLMs is the dimension of the submanifold on which they live. In Euclidean data this is usually measured by the effective dimension defined through the eigenvalues $\{\lambda_1, \lambda_2, ..\}$ of the covariance matrix of the data or

$$\lambda_{eff} = \frac{(\sum \lambda_k)^2}{\sum \lambda_k^2} . \tag{6}$$

This procedure can be extended to curved data through tangent PCA (Abboud et al., 2020) where one first uses the log map to project all data to the tangent space of the Fréchet mean by $z_i = \log_{p_F}(y_i) = \frac{d(y_i, p_F)}{\sin[d(y_i, p_F)]}(y_i - \cos[d(y_i, p_F)]p_F)$ and afterwards applies PCA in this tangent space. A large effective dimension implies that PLM representations explore many different variations in shape, whereas a small effective dimension implies that a few specific variations are enough to describe the differences in PLM representation shapes. Judging by Figure 3, it seems like PLMs go through two regimes – a dimension expansion in the first layers followed by dimension contraction towards the end. Larger models expand the dimension much more than the three-dimensional structure baseline, whereas the smaller models stay a bit closer to it. The models even show a second peak in dimension expansion.

It is also worth pointing out that, especially in later layers, this dimensionality is very low and severely different from the dimensionality found through standard PCA on the flattened PLM representation (see Figure 5 in the Appendix). We interpret this to mean that while shapes are encoded in a high-dimensional ambient space, the differences in shapes of PLM representations can be described by just a few shape descriptors. In other words, PLMs encode proteins by similar shapes

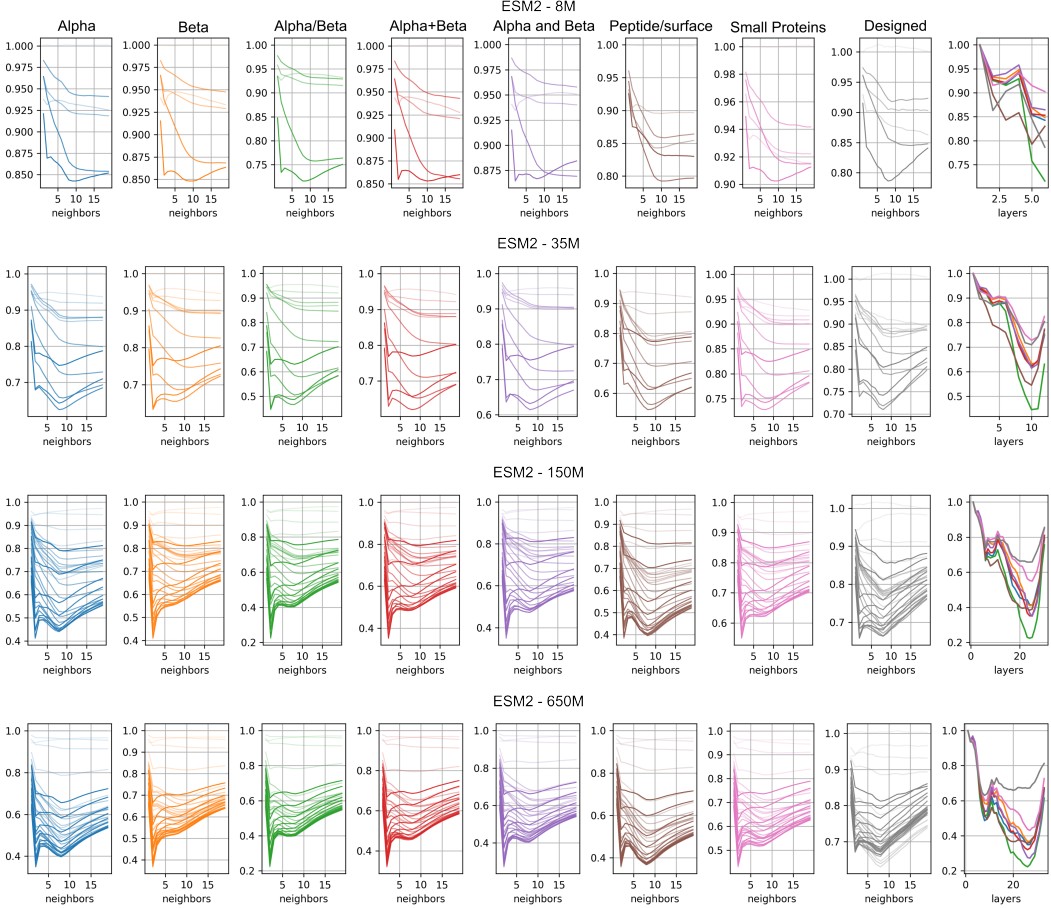

Figure 4: Graph filtration moments for all models and all layers evaluated on different protein classes from the SCOPe database. Curves increase in transparency with increasing layer number. The rightmost plots show the minimal value of the filtration as a function of the layer. Each curve is color-coded for protein class in correspondence with the other plots.

while spanning many different directions within their ambient space. To see if this trend only occurs in the ESM2 models, we also ran the same analysis on the general-purpose Ankh model Elnaggar et al. (2023). As one can see in Figure 7, a similar pattern can be seen in that model, though later layers reduce the dimension more severely.

## 3.2 CONTEXT LENGTH SENSITIVITY OF PLMS

While it is interesting to understand the global geometric features of PLM representation shape spaces, language models are known to encode contextual features of text which are hard to relate to the aforementioned measures. In order to better understand how structural context is encoded in PLMs, we use the graph filtration moment defined in Equation 4. This measure indicates how similar the shapes of PLM representations are to their three-dimensional protein structure counterparts. A value of 1 or above means that the PLM arranges all residues in a random point cloud, whereas lower values imply that three-dimensional protein structure is encoded in the PLM representation.

In Figure 4 we show the graph filtration moments across all layers of the different ESM2 models on the SCOPe protein classes. The curves for shallow layers are more transparent than deep layers and the rightmost plot shows the minimum value across the filtration as a function of depth. Several peculiar patterns arise throughout the filtration. The first thing to note is that larger models have a better similarity to three-dimensional protein structure in their intermediate layers, but model size seems to have less of an impact at the last layer. This implies that while encoding protein structure

emerges as an important intermediate processing step for unmasking, it is not as important at the last classification step.

The second striking observation is that in all models, many of the curves have a bimodal shape throughout the filtration. This implies that PLM representations encode three-dimensional protein structure at both a very local level, at about two neighbors, as well as at a slightly less local level, at about eight neighbors. The improved encoding at two neighborhoods has a simple interpretation as PLM representations having similar immediate neighbors to three-dimensional protein structure. However, the second valley at around eight neighbors is harder to interpret. Given the fact that it is less pronounced in the Beta class, one might speculate that it might be related to representations of Alpha helices, but more work is needed to understand the precise meaning of this feature.

Finally, it is also clear that certain protein classes like Alpha/Beta are represented by shapes that are much more structurally similar to their three-dimensional structure compared to other classes such as small and designed proteins. Furthermore, while initial layers sometimes showed a correlation between protein length and the minimum graph filtration moment, later layers, especially in larger models, did not show such a pattern (see Figure 9 and Figure 10). This indicates that local context structure is represented independently of protein length. The results for the Ankh model show a similar pattern as can be seen in Figure 8.

## 4    DISCUSSION

We have applied two approaches in order to better understand the geometry of shape spaces generated by PLM representations. The first uses SRV representations and quotients out rotations to create a space with Riemannian structure, which allows us to define a metric and generalize statistical methods such as PCA to shapes. The second uses graph filtrations to study how protein structure is encoded in the layers of a PLM at as many levels of resolution as one desires. Given the abstract nature of our results, here we provide a discussion of how they can be understood more intuitively and propose several future directions that would be exciting to pursue.

### 4.1    EXPANSION AND CONTRACTION IN PLM REPRESENTATION SHAPE SPACES

As shown in Figure 3, the initial layers in PLMs exhibit an expansion in effective dimensionality, whereas later layers contract the shape space to a low-dimensional subspace. Previous work on traditional language models has used the notion of intrinsic dimensionality (Li et al., 2018) and has shown that language models have a remarkably low intrinsic dimensionality relative to model size (Aghajanyan et al., 2020). Additionally, while our results concern collections of many proteins, they complement the analysis of Hakim et al. (2025), which uses IsoScore (Rudman et al., 2022) to conclude that PLM representations of individual proteins span 2-14 dimensions. Similar measures of dimensionality are also thought to relate to task performance (Marbut et al., 2024), training convergence and generalization (Ruppik et al., 2025).

The specific expansion-contraction pattern observed in our estimate of dimensionality is very similar to the behavior seen in Cheng et al. (2024) and Valeriani et al. (2023). The universal appearance of this pattern is thought to correspond to a general high-abstraction regime in the dimension-expansion phase and a specific semantically rich regime in the contraction phase. These properties of language model layers can be effectively used to solve any task.

Our approach looks at the dimensionality of the data within the shape space manifold rather than directly looking at all residue representations in the embedding space. This leads to an arguably more clear interpretation, the initial layers of a PLM represent proteins by shapes that can be flexibly deformed to each other by combining many different non-linear shape transformations. The higher the dimensionality, the more such transformations there are. Therefore, the sharp reduction in dimensionality in later layers means that there are only a few transformations that are needed to efficiently navigate PLM representation shape spaces. Understanding the precise nature of these transformations would be an exciting direction for future work.

## 4.2 STRUCTURAL ENCODING IN PLM REPRESENTATIONS

In addition to studying the geometry of PLM representation shape spaces, we also looked at how three-dimensional protein structure is encoded in the layers of a PLM by using graph filtrations. We observe a bimodal pattern in which protein structure is optimally encoded at the resolution of very short context lengths of about two residues as well as at slightly longer context lengths of around eight residues. The PLM representations of Alpha/Beta proteins showed by far the highest similarity to their three-dimensional structure, whereas small and designed proteins were represented by more distinct shapes. While outside of the scope of this work, understanding what features of these protein classes determine how much of the structure is encoded by PLMs is an exciting direction for future research.

The finding that PLMs encode three-dimensional protein structure at all is surprising given that PLMs are given masked sequences and are then trained to predict the most likely missing amino acids. There is no point at which PLM representations are incentivized to encode protein structure. Therefore, either learning protein structure is beneficial for the unmasking process or the function learned during unmasking shares some properties (one might say "correlates") with the function used in the folding of a sequence to its three-dimensional structure.

Finally, our findings further explain why folding models such as ESMFold benefit from starting with a pretrained PLM that has already partially learned protein structure. Our observation that structure is not optimally encoded in the last layer, but rather in the layers that immediately precede it, can be used to improve initializations for protein models. Given our results, we expect that better folding performance can be achieved if one uses the layers with the optimal structural PLM representation rather than the whole model. Initial attempts to show this with linear models or small networks failed to generalize (data not shown) and verifying this hypothesis would require training and testing larger models. The way in which layerwise representations can be used for folding, along with other functional tasks, is another avenue of future research that would be exciting to explore.

## 5 REPRODUCIBILITY STATEMENT

Our analysis uses publicly available datasets and protein language models. The code used to sample the data (along with a list of .cif identifiers) and to perform the analysis can be found at `https://github.com/KBeshkov/ProtGeom/`.

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

# A APPENDIX

## A.1 ADDITIONAL FIGURES

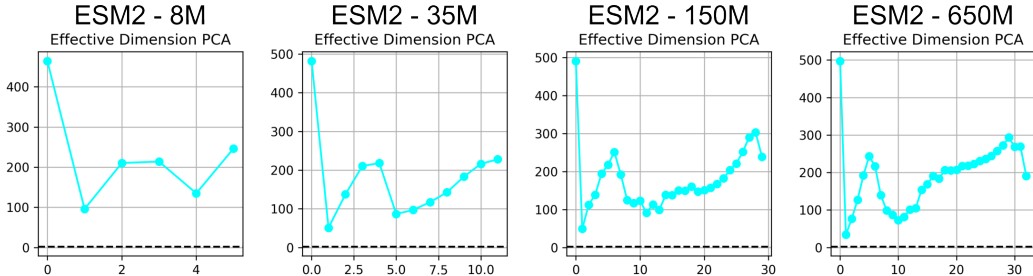

Figure 5: Effective dimension esimated using PCA directly on PLM representations. Each PLM representation is first interpolated at 1000 points. Afterwards each tensor of shape 1000×*PLM dimension at a layer* is flattened, meaning that the maximum dimension of this space is 1000 times the dimension at the layer.

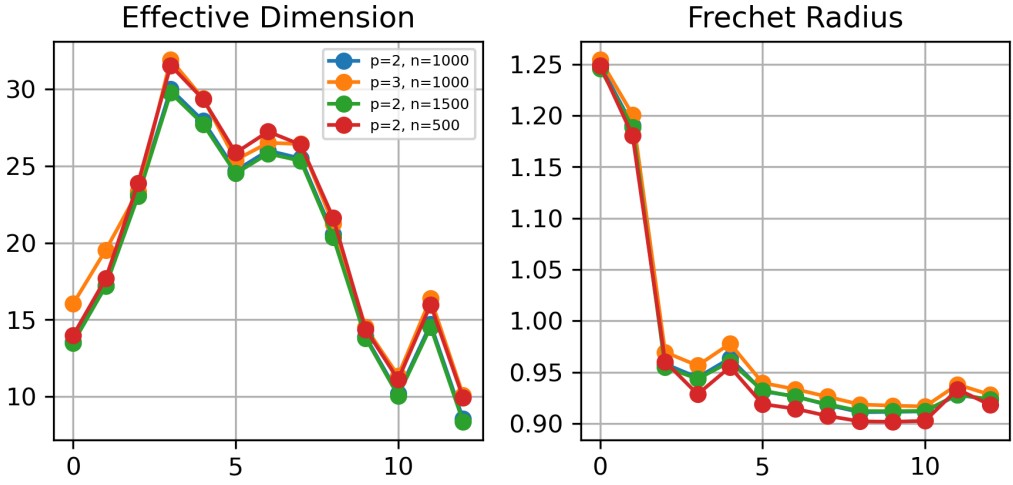

Figure 6: Robustness with respect to interpolation order (indicated by p) and number of sampled points (indicated by n) evaluated on the ESM2 - 35M model.

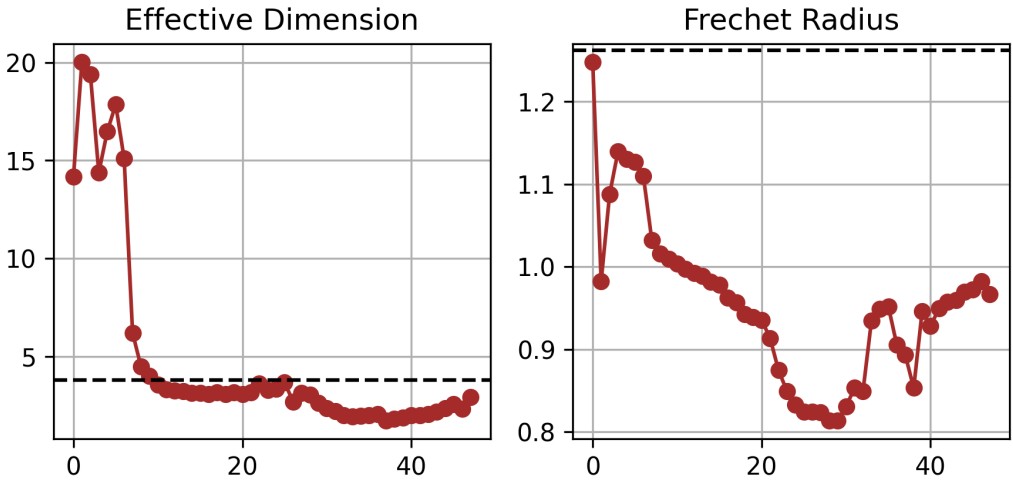

Figure 7: Effective dimension and Fréchet radius for the base Ankh model as a function of the layers.

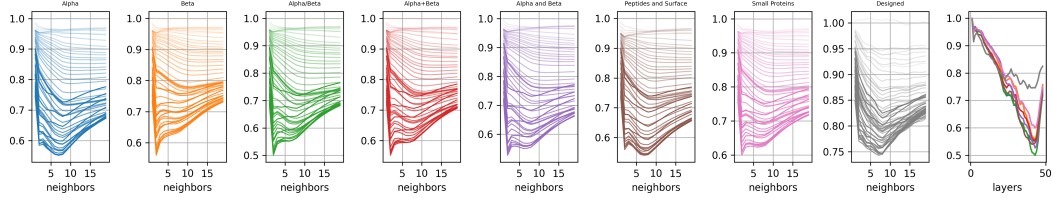

Figure 8: Graph filtration moments for the base Ankh model.

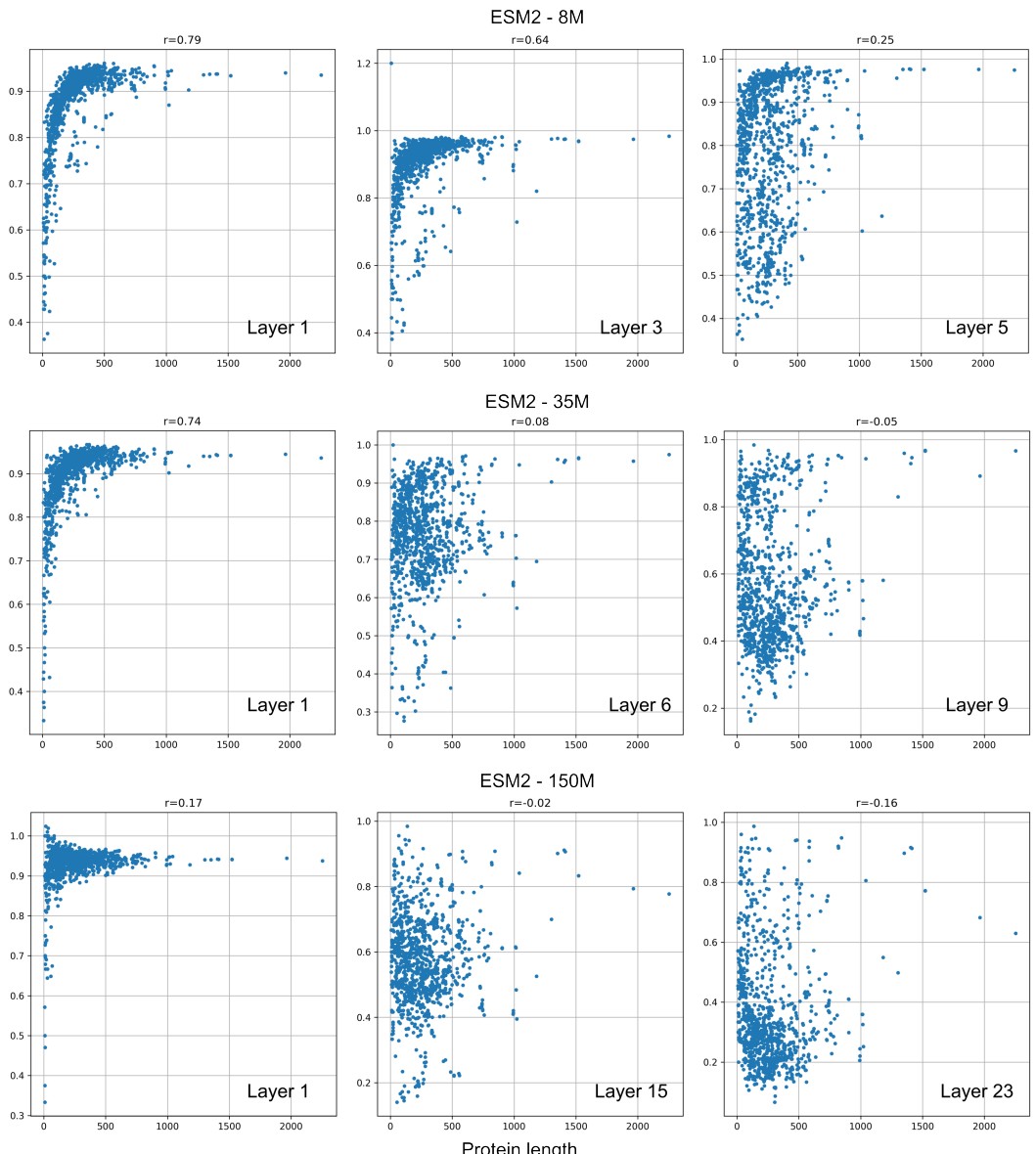

Figure 9: Protein length vs minimum graph filtration moment for several layers of the first three ESM models. Values above the plots indicate the Spearman correlation values. The last layer is chosen to correspond to the minimum in the plots in Figure 4.

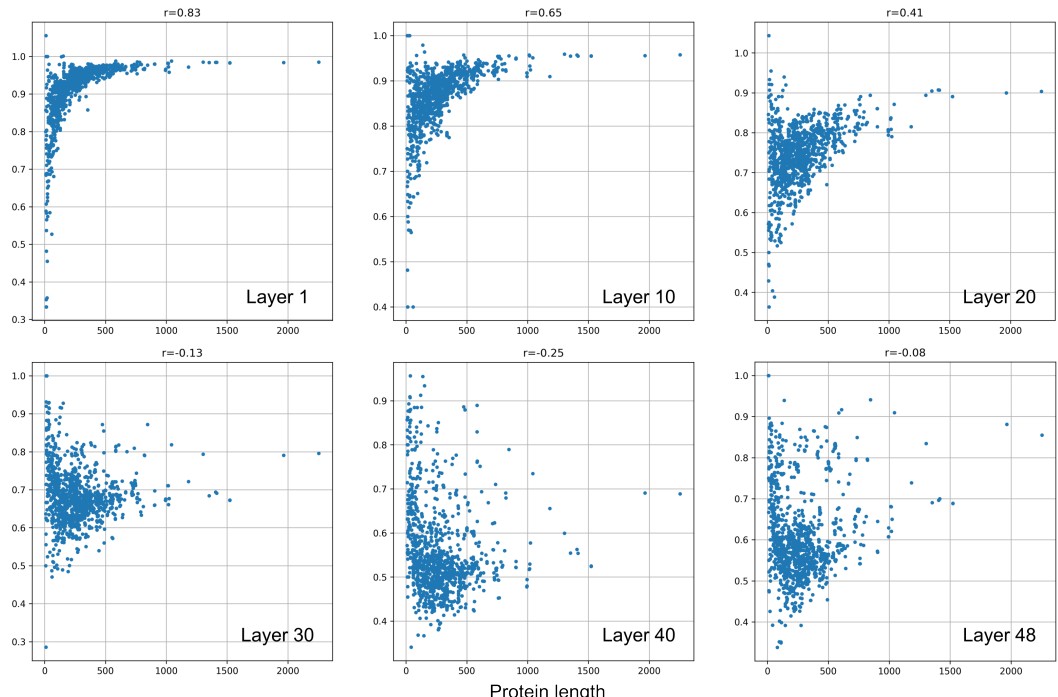

Figure 10: Protein length vs minimum graph filtration moment for several layers of the Ankh model. Values above the plots indicate the Spearman correlation values.

## A.2 ERRATUM

In a previous version of the manuscript, we had included the embedding of the beginning of sequence (BOS) token in the analysis of ESM2 and the end of sequence (EOS) token in the analysis of Ankh shape space geometry. The embeddings of these tokens tend to have much larger norm and are outliers. While this had no significant impact on our context sensitivity results, it had a noteworthy impact on the shape space analysis. Given that these tokens have no biological meaning we have decided to exclude them and update the figures in the text and the appendix. The expansion-contraction pattern in the effective dimension remains, but the absolute values for the effective dimension increased, showing the importance of excluding such tokens in future analysis. The old results including the sequence boundary tokens are shown in figures 11 and 12 as a historical record.

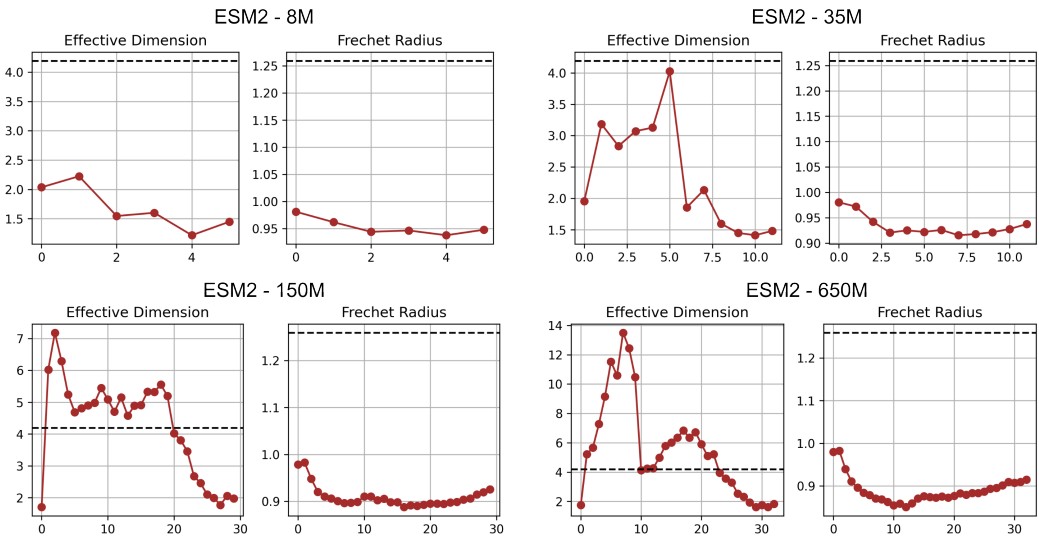

Figure 11: Old results for the ESM2 models including the BOS token

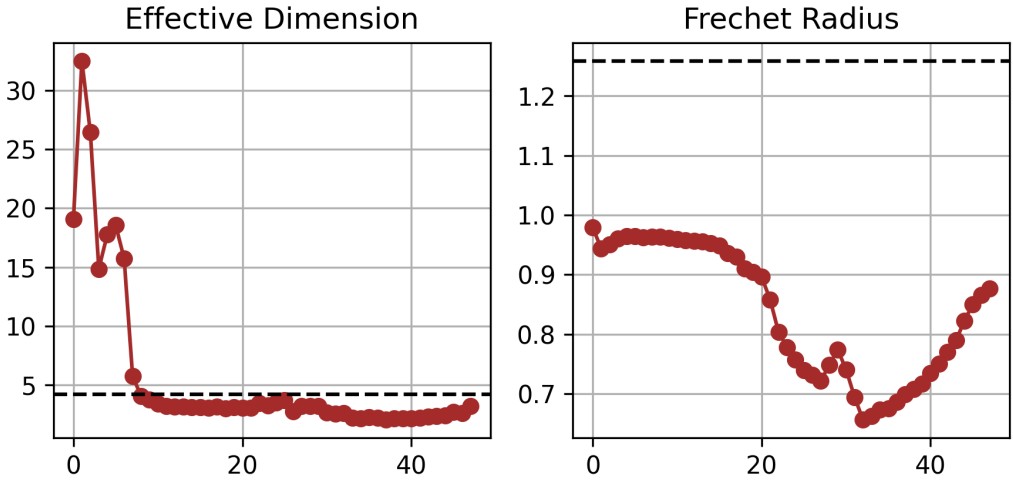

Figure 12: Old results for the Ankh model including the EOS token.

