# OpenReview forum: "Towards Understanding the Shape of Representations in Protein Language Models"
_ICLR.cc/2026/Conference — ICLR 2026 Poster_

### Official Review · Reviewer_eXtu · 2025-10-28

**Soundness:** 3
**Presentation:** 3
**Contribution:** 3
**Rating:** 6
**Confidence:** 4

**Summary:**

This paper investigates the geometric and structural properties of representations learned by PLMs, specifically focusing on ESM2 models. The authors propose two approaches: (1) a shape-space analysis using Square-Root Velocity (SRV) representations to compare protein structures and PLM embeddings in a rotation-invariant Riemannian space, and (2) a graph filtration method to analyze the context length at which PLMs encode protein structural information. The main findings include: (1) PLM representation shape spaces exhibit an expansion-contraction pattern in effective dimensionality across layers. (2)PLMs encode 3D protein structure most faithfully at very short (2 residues) and slightly longer (8 residues) context lengths. (3)The optimal structural encoding occurs in intermediate layers, not the final layer, suggesting potential improvements for folding models. The authors argue that their approach provides a novel way to understand how PLMs transform the space of protein sequences and encode structural information without explicit supervision.

**Strengths:**

- The paper introduces a novel application of shape analysis (SRV) and graph filtrations to study PLM representations. This is a fresh perspective compared to existing interpretability methods that focus on individual sequences.
- The SRV and graph filtration approaches are well-motivated and technically sound, providing a principled way to compare protein structures and embeddings.
- The expansion-contraction pattern in effective dimension and the bimodal structural encoding at specific context lengths are intriguing, which facilitates understanding of the context lengths at which models encode the structural features of proteins.
- The findings have practical implications for improving protein folding models and protein design, e.g., by selecting intermediate layers for downstream tasks.
- The paper is generally well-written, with clear motivations and a logical flow. The figures help illustrate the methods and results effectively.

**Weaknesses:**

- The study is restricted to ESM2 models and eight classes of the SCOP dataset. It would be beneficial to include other PLMs and datasets to assess generalizability.
- The paper lacks comparisons with simpler baselines (e.g., random embeddings, non-PLM representations) to better contextualize the reported metrics (e.g., Fréchet radius, effective dimension).
- While the graph filtration method is innovative, the interpretation of the “graph filtration moment” is somewhat abstract. More intuition or a qualitative analysis (e.g., visualizing which residues are captured at different context lengths) would strengthen the claims.
- The paper implies that encoding protein structure emerges as an important intermediate processing step for unmasking, but no causal experiments are provided to support this.
- The impact of interpolation methods (quadratic splines) and SRV parameterization choices is not explored.
- No statistical tests or confidence intervals are reported for the Fréchet radius or effective dimension estimates.
- No implementation details or hyperparameters are provided in the current draft. While the ESM2 model and SCOP dataset are publicly available, releasing the code of this work would greatly improve reproducibility. Although some scripts appear to be included in the supplementary .zip file, it is recommended that the authors formally release a public code repository link in the paper.

**Questions:**

- How sensitive are the SRV and graph filtration results to the choice of interpolation method (quadratic splines) and the number of interpolation points?
- Could the authors provide a qualitative analysis or visualization of the structural features captured at the 2- and 8-residue context lengths, and discuss whether the observed bimodal context-length sensitivity is associated with specific secondary structure motifs?
- Have the authors experimented with using the identified optimal layers to initialize a folding model, did it improve performance over using the final layer?
- How do the results generalize to other PLM architectures or training regimes (e.g., models trained with different objectives or data)?
- The effective dimension is computed via tangent PCA. How does this compare to intrinsic dimension estimators (e.g., MLE, TWO-NN) applied directly to the PLM embeddings?

---

> ### Author Response · Authors · 2025-11-24
>
> We wish to thank the reviewer for their time and the thorough review. We have now updated our manuscript with an extended discussion of the robustness analysis of our interpolation method along with empirical validation. We have also added analysis of the SotA Ankh PLM which leads to similar conclusions, please see the updated pdf. Below we summarize and address each question posed by the reviewer.
>
> > How sensitive are the SRV and graph filtration results to the choice of interpolation method (quadratic splines) and the number of interpolation points?
>
> The graph filtration results are independent of both as no interpolation is done for them. Representations in PLMs and 3d protein structure, require some form of interpolation to be compared. Additionally, the SRV representation requires taking a derivative which means that the curves we study have to be once differentiable. This excludes linear interpolation. Quadratic splines were therefore chosen as the simplest form of interpolation that is once differentiable. Furthermore, interpolating with high-order splines could potentially add additional structure between points on the curves (see Runge’s phenomenon) that reflect the spline rather than anything related to proteins or protein representations.
> We now discuss this in the text (lines 255-260) and have added results using cubic splines and different numbers of samples (Figure 6 in the Appendix). These results show that the effective dimension maintains the same pattern, but is slightly raised for higher order splines and low sampling, reflecting the additional structure induced by the higher order spline in sparse point clouds.
>
> > Could the authors provide a qualitative analysis or visualization of the structural features captured at the 2- and 8-residue context lengths, and discuss whether the observed bimodal context-length sensitivity is associated with specific secondary structure motifs?
>
> The graph shown at the top of Figure 2, shows an example of the filtration at different context lengths, but it does not identify particular features. Structural features at the 2-residue context are simply the local neighbors of each residue. Features at the 8-residue context length are harder to understand. In the breakdown by protein class, the Beta class has a less pronounced valley. This hints towards alpha helixes potentially playing a role in the emergence of this pattern. Given our current analysis we cannot confidently conclude that 8-context lengths encode alpha helixes, but we now mention this potential feature in the results (lines 374-377).
>
> > Have the authors experimented with using the identified optimal layers to initialize a folding model, did it improve performance over using the final layer?
>
> While we do not have the resources to train a model on the scale of ESM-Fold, we have attempted to train linear as well as three layer networks as a folding head on top of the pre-last layer PLM representations. So far we have not been successful in generalizing with such small networks as they essentially overfit on all layers except the very first few. We now state this in the discussion.

---

> > ### Author Response · Authors · 2025-11-24
> >
> > >How do the results generalize to other PLM architectures or training regimes (e.g., models trained with different objectives or data)?
> >
> > To address this question, we have now added the same analysis on the hidden layers of the base Ankh model which is a SotA PLM used to predict folding, contact maps, secondary structure and others. Our results can be seen in Figures 7 & 8 and show a similar pattern to that of the ESM2 models in both effective dimension and graph filtration moment.
> >
> > > The effective dimension is computed via tangent PCA. How does this compare to intrinsic dimension estimators (e.g., MLE, TWO-NN) applied directly to the PLM embeddings?
> >
> > We discuss the relation between our approach and intrinsic dimensionality in section 4.1. The takeaway is that both approaches identify a dimension expansion in the early layers, though our approach shows a second expansion phase in larger models.
> >
> > What is important to keep in mind is that these correspond to dimensionalities of different spaces. The intrinsic dimension is the dimension of the manifold on which all points of the PLM representation live and is bounded by the width of the observed layer. The effective dimension that we look at is the dimension of the manifold in shape space. The shape space of all curves is infinite dimensional (though in practice it will be bounded by the number of interpolation points = 1000 which is way higher than what we observe as effective dimension) and the curves on that space form a low-dimensional submanifold.
> >
> > > There is a lack of statistical verification for the Frechet radius and the effective dimensionality. There is a lack of implementation details and hyperparameters in the current manuscript.
> >
> > We agree that providing statistical verification is a very important next step and we are working towards it. Unfortunately with our current resources estimating one effective dimensionality curve can take up to a week for the largest model (ESM-650M). Repeating this estimation enough times to provide proper confidence intervals and statistics is therefore unfeasible for us at this stage. We have provided a .zip file of our code to maintain anonymity but will publicly release a full repository with our code upon potential acceptance.
> >
> > We hope that our analysis of the robustness to interpolation parameters as well as the additional analysis on the Ankh model and the extended discussion and results sections have managed to address the reviewers' concerns. We welcome any additional comments, questions or suggestions.

---

### Official Review · Reviewer_Xyzz · 2025-10-31

**Soundness:** 3
**Presentation:** 4
**Contribution:** 4
**Rating:** 8
**Confidence:** 4

**Summary:**

The authors compare spaces induced by protein language models to those induced by protein structures. By applying various representation schemes and dimensionality analysis techniques they derive some insights about the way PLM embeddings are organized. In particular, we learn that protein structure is mirrored in PLM embeddings at highly localized neighbourhoods, and that the effective shape space of embeddings is quite limited. These are highly valuable insights for future PLM model development.

**Strengths:**

* This work provides a highly unique and novel view on PLM work and generates valuable insights. The framework for understanding PLMs is also novel and will be a valuable asset for the field.
* Modeling the 'shape' of embeddings is a very interesting proposition. I am also very surprised by the finding that embedding shape can mirror protein shape at all.
* The paper is very well-written and flows nicely from the initial premise and questions to be explored (e.g. "If structure determines function, then it is reasonable to assume that similar structure determines similar function. Furthermore, if protein representations in PLMs characterize structural, evolutionary and functional features, then one might expect that similar representations share such features. ")

**Weaknesses:**

* There is a lack of simpler methods than the one proposed. The authors begin by asking whether PLM embeddings for structurally similar proteins would be similar themselves. Of course this can be tackled by mapping the embeddings to shape spaces and comparing shapes, but one could also try to directly measure correlation for example between TM scores and PLM embeddings.

**Questions:**

* How sensitive is the analysis to the curve fitting degree? How do you measure the quality of a curve fitting for a given protein? As far as I know, this representation approach is not well-established for proteins so its soundness should be explored further.

---

> ### Author Response · Authors · 2025-11-24
>
> We wish to thank the reviewer for their time and the encouraging review. Below we address the question of interpolation degree.
>
> > How sensitive is the analysis to the curve fitting degree? How do you measure the quality of a curve fitting for a given protein? As far as I know, this representation approach is not well-established for proteins so its soundness should be explored further.
>
> It is true that this is a non-standard approach for proteins. We have now extended the text with additional discussion of why our choice of quadratic interpolation is justified. We have also performed simulations validating the robustness of this approach on the ESM2-35M model. These updates can be found in the newly uploaded version of our manuscript (lines 255-260 and Figure 6 in the Appendix).
>
> Essentially, there is no a priori truth about how one should connect two residues or their PLM representations. However, the higher the degree, the more assumptions one makes about the structure of this connection. We have therefore taken the simplest curve that remains differentiable as that property is necessary to take derivatives as required by the SRV representation in equation 2. One could choose a higher-degree spline, but that would amount to adding more structure that is due to the spline rather than in the data itself. It might also lead to artificial “wiggles” or oscillations between distance points on the curve (similar to Runge’s phenomenon for polynomial interpolation).
>
> We wish to thank the reviewer once again and welcome any further questions or suggestions.

---

### Official Review · Reviewer_R8RE · 2025-11-01

**Soundness:** 3
**Presentation:** 2
**Contribution:** 3
**Rating:** 4
**Confidence:** 4

**Summary:**

This paper aims to unravel the geometry of hidden representations in Protein Language Models (PLMs), a critical gap in understanding how PLMs encode protein structure/function for de novo design. Using two novel analytical frameworks—square-root velocity (SRV) shape spaces and graph filtrations—the authors analyze multiple sizes of ESM2 (a leading PLM) on 1098 proteins from the SCOP dataset (spanning 8 protein classes). Key findings include: (1) PLM representation shape spaces exhibit non-linear patterns in Karcher mean and effective dimension across layers, with larger models showing "dimension expansion" in early layers and "contraction" in later layers; (2) PLMs optimally encode protein structure at short context lengths (~2 amino acids) and moderately short lengths (~8 amino acids), but degrade at larger contexts; (3) structural encoding is strongest in layers just before the last layer, suggesting potential improvements for folding models.

**Strengths:**

1. Adapts shape analysis (SRV) and topological data analysis (graph filtrations)—previously used for 3D protein structure—to PLM representation spaces, enabling rigorous metric comparisons between protein structures and their PLM embeddings.
2 Identifies a universal "expansion-contraction" pattern in effective dimension across ESM2 layers, where early layers explore diverse shape variations and later layers converge to a low-dimensional subspace of biologically relevant shapes.
3. Quantifies that PLMs prioritize local residue relationships (contexts of 2 or 8 amino acids) over global ones, explaining why PLMs implicitly encode structural features despite being trained only on masked sequence prediction.
4. Highlights that the last layer of PLMs is not optimal for structural encoding; using pre-last layers could improve downstream folding models (e.g., ESMFold).

**Weaknesses:**

1. The study only uses ESM2. It is unclear if the "expansion-contraction" pattern or context length bimodality (2, 8) applies to other popular PLMs (e.g., ProtTrans, AlphaFold’s embedding layers) or PLMs trained on different datasets (e.g., UniProt vs. Pfam).
2 The authors use Euclidean distance for graph filtrations but do not test if alternative metrics (e.g., cosine similarity, Manhattan distance) alter the bimodal context pattern. Euclidean distance may not be optimal for high-dimensional PLM embeddings, where cosine similarity is often preferred.
3. The paper suggests using pre-last layers to improve folding models but provides no experimental evidence (e.g., comparing ESMFold performance with pre-last vs. last layers). This makes the suggestion speculative. It is interesting to see such findings, and I am also curious whether the pre-last layers can improve other downstream tasks, like GO, EC prediction?
4. The paper notes that shape space effective dimension is much lower than flattened PLM representations but does not identify which specific shape features (e.g., helix curvature, beta-sheet spacing) drive this low dimensionality.
5. Most of the findings or conclusions of this theory are conforming to cognition.

**Questions:**

Does sequence length correlate with structural encoding quality?
The authors find PLMs preferentially encode immediate as well as local relations between residues, but start to degrade for larger context lengths, does this mean that PLMs cannot learn global relations?

---

> ### Author Response · Authors · 2025-11-24
>
> We wish to thank the reviewer for their time and the valuable comments regarding our manuscript. We have updated our results with simulations on an additional model and a better verification of our approach. Please see our newly updated pdf along with our responses below.
>
> **(W1)** The study only explores the ESM2 models and it is unclear what the results would be for other protein language models.
>
> To address this point, we have now extended our analysis to the SOTA Ankh base model, which is a successor of the ProtTrans model. Our newly updated results (Figures 7 and 8) show that the pattern in the effective dimensionality and the graph filtration moments in this model is highly similar to that in the ESM2 models. This suggests that the discovered expansion-contraction pattern and the bimodal sensitivity to encoding lengths is a more general phenomenon than we previously suggested.
>
> **(W2)** The study uses the Euclidean distance for graph filtrations instead of the cosine similarity or other metrics.
>
> Our analysis of context length in terms of graph filtrations is essentially a filtration of contact maps with different thresholds. While we agree that metrics such as the cosine similarity are a reasonable choice for understanding the embedding space in language models, they are a less natural choice for actual proteins where contact maps are computed using Euclidean distances. Given that the graph filtration approach always involves a comparison between real proteins and embeddings we have omitted this analysis.
>
> **(W3)** The study suggests that pre-last layers of PLMs would lead to better performance on folding tasks, but it does test folding performance.
>
> We agree that such a result would make our conclusions much stronger. Our initial attempts of training a simple linear or three layer folding head that predicts the distance matrix between residues have so far failed to generalize well in all layers. Therefore, it seems that more expensive architectures would be needed to prove this point. Unfortunately, training folding models on par with ESMFold is beyond our current resources, but we hope we can illuminate the role of the structure in pre-last layers in future work. We now point to this limitation of our work in the discussion. We agree that other downstream tasks like GO and EC prediction are also worth exploring and have updated the discussion.
>
> **(W4)** The work does not identify which shape features drive the low dimensionality.
>
> It is unclear whether structural properties such as alpha helixes and beta sheets can be considered to exist in PLM representations, which makes answering this question difficult. Additionally, shape spaces consider the global shape of each protein so the features driving the dimensionality have to be some non-linear combination of local features (like helices or sheets or another shape pattern). While this is a very exciting direction, exploring this in depth would require the development of novel tools that can consider such more local features. We thank the reviewer for this important suggestion and hope to explore it further in future work.
>
> **(Q1)** Does sequence length correlate with structural encoding quality?
>
> Our results in Figure 4 support the idea that longer relations are less well represented in PLMs. We have now also plotted the minimum graph filtration moment as a function of sequence length for a few layers in the Ankh model (see Figure 9 in the Appendix, we are in the process of running these simulations for the ESM models as well). There is a weak correlation between the two in the first layers and no strong correlation between them in later layers. This means that local relations are equally well represented in short and long proteins. However, one should keep in mind that since longer proteins contain more long-range relations, which are not well represented by PLMs, overall we would expect that more of their relations will be inconsistently represented.
>
> **(Q2)** The authors find PLMs preferentially encode immediate as well as local relations between residues, but start to degrade for larger context lengths, does this mean that PLMs cannot learn global relations?
>
> While it is possible that it is the case that PLMs cannot learn global relations, our analysis does not clearly conclude that. It is possible that PLMs generally fail to encode general unimportant global relations, but succeed in encoding key long-range contacts. Answering this specific question would require repeating our analysis on datasets specifically filtered for such examples.
>
>
> We hope that with this response, as well as the updated text, we have managed to address the reviewer's concerns and are excited to discuss further. Please let us know if you have any further suggestions or questions.

---

> ### Comment · Reviewer_R8RE · 2025-11-26
>
> Thanks for the rebuttal. It is hoped that the author can present more results and conclusions on this issue in the future.

---

> > ### Author Response · Authors · 2025-12-02
> >
> > We are happy to see that our rebbutal improved your view of our paper. Thank you for your suggestions regarding folding/GO/EC prediction as well as extracting the features that drive low dimensionality. We are very excited to explore them further in future work.

---

### Author Response · Authors · 2025-12-03

We wish to thank all reviewers for their positive, constructive and valuable feedback one last time. It is unfortunate that due to the recent events with Openreview, we did not manage to have a thorough back and forth discussion. We wish to take the opportunity to summarize how our manuscript has evolved as a result of the discussion phase and hope that this summary can be useful to new readers and AC’s. We list all changes below.

- We have now extended our analysis beyond the ESM2 models and have also studied the shape spaces and graph filtrations of the base Ankh models. The Ankh model exhibits a similar expansion-contraction pattern in the effective dimension and a bimodal curve in the graph filtration moment in its deeper layers. In addition, it achieves the highest structural similarity to 3d structure in the pre-last layers just as in the ESM2 models. This supports the idea that the patterns we identify might be more universal than just the ESM2 models.

- We extended the discussion for our choice of quadratic interpolation and have verified how the choice of interpolation degree or sampling density impacts our results. While our method is qualitatively robust to changes in the degree or sampling density, there is an expected bias in the effective diemension as the degree increases. This is not surprising given the additional complexity introduced in higher-order interpolation which reflects the method of interpolation more than the data iteself.

- We have evaluated how the graph filtration moment relates to the protein length, showing that for deeper layers that produce more structurally similar representations, the two are not related. This provides evidence that local relations are preserved in PLM representations regardless of the size of the protein.

- We have extended the results and discussion sections in order to further motivate the additions above.

We are excited to continue working on understanding the shape of PLM representations and their impact on biologically relevant tasks, as also suggested by the reviewers. We believe these changes have significantly improved the quality of our paper and hope that our readers enojy our work more as a result.

---

### Public Comment · ~Kosio_Beshkov1 · 2026-05-12
**Final camera ready version and correction**

We wish to thank the ICLR2026 organizing team for giving us the opportunity to update the camera ready version of the manuscript.

In our previous analysis we had included the ESM2 beginning (BOS) and Ankh end of sequence (EOS) tokens. Given that these tokens do not have any biological meaning and their embeddings are outliers compared to residue embeddings, we have decided to exclude them from the main analysis. We have therefore corrected the text, added an Erratum section in the Appendix in which we show the results including these tokens, and updated the main figures in the text to show the results purely based on residue embeddings. While this did not have a noticeable impact on the context sensitivity results, it did impact the shape space results. This is especially visible in the effective dimensionality figures, which maintain the general contraction-expansion pattern, but change their local shape and go up in absolute value.

---

### Meta-Review · Area_Chair_Uuin · 2026-01-05

**Summary:**

The primary reservations expressed by the reviewers focused on the study's empirical scope, methodological robustness, and the lack of experimental validation for its practical claims. A recurrent critique was the initial limitation of the analysis to the ESM2 architecture, which raised concerns regarding the generalizability of the observed expansion-contraction patterns to other PLMs, although the authors attempted to address this by adding Ankh model data during the rebuttal. Reviewers also scrutinized the technical rigor of the proposed analytical frameworks, specifically questioning the sensitivity of the results to arbitrary choices such as interpolation spline degrees and distance metrics, while noting a distinct absence of statistical confidence intervals and simple baselines to contextualize the complex topological metrics. The reviewers highlighted a gap between the paper's theoretical insights and their application, pointing out that the recommendation to utilize pre-last layers for downstream tasks like protein folding remained speculative and unsupported by direct causal experiments or performance benchmarks against standard models.

**Reviewer Concerns:**

The rebuttal successfully mitigated concerns regarding the study's generalizability. The authors effectively addressed methodological questions about the sensitivity of their shape space analysis by providing new simulations that verified stability across different interpolation degrees and sampling densities.

**Reviewer Scores:**

Based on the trajectory of the rebuttal, it is highly probable that the reviewers would have maintained their respective scores of 6, 6, and 8. Reviewer R8RE, initially the most skeptical at a 4, would almost certainly have upgraded to a 6, as the inclusion of the Ankh model and the theoretical justification for Euclidean distance directly resolved their primary complaints regarding generalizability and metric suitability. Reviewer eXtu would likely have maintained their score of 6. Reviewer Xyzz would have steadfastly retained their score of 8.

---

### Decision · Program_Chairs · 2026-01-26

Accept (Poster)